# Proteogenomic Profiling of Treatment-Naïve Metastatic Malignant Melanoma

**DOI:** 10.3390/cancers17050832

**Published:** 2025-02-27

**Authors:** Magdalena Kuras, Lazaro Hiram Betancourt, Runyu Hong, Leticia Szadai, Jimmy Rodriguez, Peter Horvatovich, Indira Pla, Jonatan Eriksson, Beáta Szeitz, Bartłomiej Deszcz, Charlotte Welinder, Yutaka Sugihara, Henrik Ekedahl, Bo Baldetorp, Christian Ingvar, Lotta Lundgren, Henrik Lindberg, Henriett Oskolas, Zsolt Horvath, Melinda Rezeli, Jeovanis Gil, Roger Appelqvist, Lajos V. Kemény, Johan Malm, Aniel Sanchez, Attila Marcell Szasz, Krzysztof Pawłowski, Elisabet Wieslander, David Fenyö, Istvan Balazs Nemeth, György Marko-Varga

**Affiliations:** 1Department of Translational Medicine, Lund University, Skåne University Hospital Malmö, 214 28 Malmö, Sweden; magdalena.kuras@bme.lth.se (M.K.); jeovanis.gil_valdes@med.lu.se (J.G.); johan.malm@med.lu.se (J.M.); aniel.sanchez@med.lu.se (A.S.); krzysztof.pawlowski@utsouthwestern.edu (K.P.); 2Department of Biomedical Engineering, Lund University, 221 00 Lund, Sweden; p.l.horvatovich@rug.nl (P.H.); indira.pla_parada@bme.lth.se (I.P.); jonatan.eriksson@axis.com (J.E.); yutaka.sugihara@med.lu.se (Y.S.); henrik.lindberg@med.lu.se (H.L.); melinda.rezeli@bme.lth.se (M.R.); roger.appelqvist@bme.lth.se (R.A.); gyorgy.marko-varga@bme.lth.se (G.M.-V.); 3Department of Clinical Sciences Lund, Division of Oncology, Lund University, 221 00 Lund, Sweden; charlotte.welinder@med.lu.se (C.W.); bo.baldetorp@med.lu.se (B.B.); lotta.lundgren@med.lu.se (L.L.); henriett.kovacsne_oskolas@med.lu.se (H.O.); 4Institute for Systems Genetics, NYU Grossman School of Medicine, New York, NY 10016, USA; rh2740@nyu.edu (R.H.); david@fenyolab.org (D.F.); 5Department of Biochemistry and Molecular Pharmacology, NYU Grossman School of Medicine, New York, NY 10016, USA; 6Department of Dermatology and Allergology, University of Szeged, 6720 Szeged, Hungary; szadai.leticia@med.u-szeged.hu (L.S.); nemeth.istvan.balazs@med.u-szeged.hu (I.B.N.); 7Department of Biochemistry and Biophysics, Karolinska Institute, 171 77 Stockholm, Sweden; jimmy.rodriguez@thermofisher.com; 8Department of Analytical Biochemistry, Faculty of Science and Engineering, University of Groningen, 9712 CP Groningen, The Netherlands; 9Division of Oncology, Department of Internal Medicine and Oncology, Semmelweis University, 1085 Budapest, Hungary; 10Department of Biochemistry and Microbiology, Warsaw University of Life Sciences, 02-787 Warsaw, Poland; bartek_deszcz@wp.pl; 11SUS University Hospital Lund, 222 42 Lund, Sweden; christian.ingvar@med.lu.se; 12Department of Surgery, Clinical Sciences, Lund University, SUS, 221 00 Lund, Sweden; 13HCEMM-SU Translational Dermatology Research Group, Semmelweis University, 1085 Budapest, Hungary; kemeny.lajos@semmelweis.hu; 14Department of Dermatology, Venereology and Dermatooncology, Faculty of Medicine, Semmelweis University, 1085 Budapest, Hungary; 15Department of Physiology, Faculty of Medicine, Semmelweis University, 1085 Budapest, Hungary; 16MTA-SE Lendület “Momentum” Dermatology Research Group, Hungarian Academy of Sciences and Semmelweis University, 1085 Budapest, Hungary; 17Department of Bioinformatics, Semmelweis University, 1085 Budapest, Hungary; szasz.attila_marcell@med.semmelweis-univ.hu; 18Department of Molecular Biology, University of Texas Southwestern Medical Center, Dallas, TX 75390, USA; 19Chemical Genomics Global Research Lab, Department of Biotechnology, College of Life Science and Biotechnology, Yonsei University, Seoul 03722, Republic of Korea; 201st Department of Surgery, Tokyo Medical University, Tokyo 160-8402, Japan

**Keywords:** melanoma, proteogenomics, proteomics, subtypes, stratification, histopathology, BRAF V600E, tumor microenvironment, single amino acid variants, lymph node metastases

## Abstract

This study investigates metastatic melanoma through an in-depth analysis of molecular and genetic characteristics of tumor samples. Five distinct melanoma subtypes were identified based on the protein expression profiles within the tumors and their surrounding microenvironments. Our findings highlight the role of specific mutations, particularly in the BRAF gene, which can affect patient survival, with some patients showing better outcomes due to a particular immune response triggered by the mutation. Additionally, our study uncovered genetic changes in melanoma tumors that could support the development of treatments that target the immune system. The study shows that proteins linked to survival are distributed in complex patterns in both primary tumors and metastases, offering deeper insights into the molecular heterogeneity of melanoma. By combining detailed molecular analysis with clinical and tissue examination, this research provides new insights into melanoma biology and offers guidance for improving treatment strategies.

## 1. Introduction

Malignant melanoma is the most aggressive skin cancer, with high metastatic potential [1,2], and is responsible for 80% of skin cancer-related deaths [3]. Due to its heterogeneous nature and unpredictable metastatic progression, melanoma poses a significant challenge to the healthcare system [4].

In recent decades, genomic studies revealed the importance of the oncogenic driver mutations in BRAF, NRAS, NF1, and KIT genes in tumor development [3,5,6,7,8]. The oncogenic mechanisms of the PTEN/phosphoinositol-3-kinase signaling pathway, RAC1, CDKN2A, telomerase reverse transcriptase promoter mutations, and several molecular alterations specific to mucosal and chronically sun-damaged melanomas have also been implicated in disease progression [9].

Recently, multiple effective treatment options have been developed for metastatic melanoma by targeting the tumor cells using kinase inhibitor therapies or targeting the surrounding tumor microenvironment (TME) by the indirect action of immune checkpoint blockade-based therapies. However, many patients do not respond to immunotherapy, and patients on targeted therapy eventually progress [10,11].

In this context, the field of omics has significantly enhanced our understanding of melanomas, shedding light on distinct aspects of tumor progression and metastasis. These developments have been predominantly driven by genomics and transcriptomic studies encompassing tumor tissues and derived cell lines [7,12,13,14]. However, the contribution of proteomic studies using actual patient tumor samples remains relatively limited. In primary melanomas, proteomic profiling has identified biomarkers associated with disease progression and patient outcomes [15,16]. Immune-related proteins, including MHC molecules, have been shown to influence tumor immune evasion and growth, providing potential targets for immunotherapy [17]. In metastatic melanoma, emerging research has highlighted the significant role of mitochondrial dependence, organ-specific heterogeneity, and sex-based differences in influencing the tumor’s response to or resistance against therapies [18,19,20,21].

Despite these advancements, a comprehensive stratification of primary melanoma beyond the established histopathological subtypes and a unified proteomic classification system for metastases remains underdeveloped and absent. Establishing such stratification has the potential to enhance diagnostic precision, uncover novel prognostic and predictive biomarkers, guide therapeutic decision-making, and identify new therapeutic targets.

Recognizing the profound heterogeneity of melanoma, as highlighted by numerous studies, we aimed to undertake a deeper proteomic investigation of melanoma metastases integrated with transcriptomics, genomics, histopathology, and patient data. In the present study, we analyzed a cohort of treatment-naïve lymph node metastases collected prior to the era of immune checkpoint inhibitors and targeted therapies. This approach distinguishes our research from others, which often center on therapy response and non-response dynamics. We hypothesized that a comprehensive proteogenomic analysis of these samples would yield a unique understanding of melanoma biology in the context of natural disease progression. The breadth and depth of our study and the complexity and heterogeneous nature of melanoma allowed us to focus on multi-faceted aspects of melanoma, characterizing and revealing novel molecular classifiers that bridge cancer cell biology, the tumor microenvironment (TME), survival markers, and the expression of the mutational genetic landscape. The analysis uncovered (1) proteomic subtypes that integrate tumor-immune and stromal components, linking these subtypes to clinical and histopathological features, (2) classifications specifically for BRAF V600-mutated metastases, and (3) subgroups defined by the TME. Additionally, we identified a landscape of melanoma-associated single amino acid variants (SAAVs) and derived potential neoantigens. In parallel, spatial expression patterns of survival-related protein markers were analyzed across both metastases and primary tumors, further emphasizing the molecular heterogeneity of melanoma.

## 2. Materials and Methods

### 2.1. Collection of Malignant Melanoma Samples and Ethical Approval

#### 2.1.1. Cohorts

Detailed information about the cohorts is provided in Appendix A. A summary of the clinical parameters of the cohort can be found in Appendix A.

#### 2.1.2. Ethical Approval

The study of the discovery cohort was approved by the Regional Ethical Committee at Lund University, Southern Sweden (DNR 191/2007, 101/2013 (BioMEL biobank), 2015/266, and 2015/618). All patients provided written informed consent. The study has been performed in compliance with GDPR.

The cohort used for immunohistochemistry was collected according to the guidelines of the Declaration of Helsinki. All patient samples were obtained with the approval of the Hungarian Ministry of Human Resources, Deputy State Secretary for National Chief Medical Officer, Department of Health Administration with the ethical authorization number MEL-PROTEO-001, 4463-6/2018/EÜIG, and the date of approval was 12 March 2018. Based on the MEL-PROTEO-001, 4463-6/2018/EÜIG ethical approval, informed consent was not applicable due to the retrospective anonymized FFPE samples.

### 2.2. Sample Preparation for Mass Spectrometry

#### 2.2.1. Protein Extraction, Digestion, and C18 Desalting

Protein extraction was performed on sectioned (30 × 10 µm) fresh-frozen melanoma tissues using the Bioruptor plus model UCD-300 (Dieagenode, Denville, NJ, USA). In total, 142 melanoma samples were lysed in 100 µL lysis buffer containing 4 M urea and 100 mM ammonium bicarbonate. After a brief vortex, samples were sonicated in the Bioruptor for 40 cycles at 4 °C. Each cycle consisted of 15 s at high power and 15 s without sonication. The samples were then centrifuged at 10,000× *g* for 10 min at 4 °C. The protein content in the supernatant was determined using a colorimetric micro–BCA Protein Assay kit (Thermo Fisher Scientific, Waltham, MA, USA).

Urea in-solution protein digestion was performed on the AssayMAP Bravo (Agilent Technologies, Santa Clara, CA, USA) micro-chromatography platform using the digestion v2.0 protocol. Protein concentrations were adjusted to 2.5 µg/µL and 100 µg of protein from each sample were reduced with 10 mM DTT for 1 h at room temperature (RT) and sequentially alkylated with 20 mM iodoacetamide for 30 min in the dark at RT [22]. The samples were diluted approximately seven times with 100 mM ammonium bicarbonate to decrease the urea concentration. Digestion was performed in two steps at RT. Proteins were first incubated with Lys-C at a 1:50 (*w*/*w*) ratio (enzyme/protein) for 5 h, and then trypsin was added at a 1:50 (*w*/*w*) ratio (enzyme/protein), and the mixture was incubated overnight. The reaction was quenched by adding 20% TFA to a final concentration of ~1%. Peptides were desalted on the AssayMAP Bravo platform using the peptide cleanup v2.0 protocol. C18 cartridges (Agilent, 5 µL bed volume) were primed with 100 µL of 90% acetonitrile (ACN) and equilibrated with 70 µL of 0.1% TFA at a flow rate of 10 µL/min. The samples were loaded at 5 µL/min, followed by an internal cartridge wash with 0.1% TFA at a flow rate of 10 µL/min. Peptides were eluted with 30 µL 80% ACN, 0.1% TFA, and dried in a Speed-Vac (Eppendorf, Hamburg, Germany) before TMT labeling.

Protein digestion and C18 peptide cleanup were repeated for the phosphoproteomic analysis on the protein lysates. After Speed-Vac, the peptides were resuspended in 80% ACN and 0.1% TFA prior to the phosphopeptide enrichment.

#### 2.2.2. TMT 11 Plex Labeling

The peptide amount in each sample was estimated using a quantitative colorimetric peptide assay kit (Thermo Fisher Scientific). Within each batch, equal amounts of peptides were labeled with TMT11 plex reagents, using a ratio of 0.8 mg reagent to 100 μg of peptides. The TMT labeling was performed according to the manufacturer’s instructions. Peptides were resuspended in 100 µL of 200 mM TEAB, and individual TMT11 plex reagents were dissolved in 41 µL of anhydrous ACN and mixed with the peptide solution. The internal reference sample, a pool of aliquots from protein lysates from 40 melanoma patient samples, was labeled in channel 126 in each batch. After one hour of incubation, the reaction was quenched by adding 8 µL of 5% hydroxylamine and incubated at room temperature for 15 min. The labeled peptides were mixed in a single tube, the volume was reduced in a Speed-Vac, and then the peptides were cleaned up using a Sep-Pak C18 96-well Plate (Waters, Milford, MA, USA). The eluted peptides were dried in a Speed-Vac and resuspended in water before high pH RP-HPLC fractionation. The samples were distributed among 15 batches, using TMT tag 126 as the internal reference sample, as described in Appendix A.

#### 2.2.3. High pH RP-HPLC Fractionation

The TMT11 batches were fractionated using an Aeris Widepore XB-C8 (3.6 μm, 2.1 × 100 mm) column (Phenomenex, Torrance, CA, USA) on an 1100 Series HPLC (Agilent, Santa Clara, CA, USA) operating at 80 µL/min. The mobile phases were solvent A, 20 mM ammonium formate pH 10, and solvent B, 80% ACN and 20% water containing 20 mM ammonium formate pH 10. An estimated amount of 200 µg was separated using the following gradient: 0 min 5% B; 1 min 20% B; 60 min 40% B; 90 min 90% B; 120 min 90% B. The column was operated at RT, and the detection wavelength was 220 nm. Then, 96 fractions were collected at 1 min intervals and further concatenated to 24 or 25 fractions (by combining 4 fractions that were 24 fractions apart so that #1, #25, #49, and #73, and so forth, were concatenated), and dried in a Speed-Vac.

#### 2.2.4. Automated Phosphopeptide Enrichment

The Phospho Enrichment v2.0 protocol on the AssayMAP Bravo platform [23] was used to enrich phosphorylated peptides using 5 µL Fe(III)-NTA cartridges. The cartridges were primed with 100 µL of 50% ACN, 0.1% TFA at a flow rate of 300 µL/min and equilibrated with 50 µL of loading buffer (80% ACN, 0.1% TFA) at 10 µL/min. Samples were loaded onto the cartridge at 3.5 µL/min. The samples were washed with 50 µL of loading buffer, and the phosphorylated peptides were eluted with 25 µL of 5% NaOH directly into 10 µL of 50% formic acid. Samples were dried in a Speed-Vac and stored at −80 °C until analysis by LC-MS/MS.

### 2.3. MS Data Acquisition

#### 2.3.1. nLC-MS/MS Analysis

The nLC-MS/MS analysis was performed on an Ultimate 3000 HPLC coupled to a Q Exactive HF-X mass spectrometer (Thermo Scientific). Each fraction (1 µg) was loaded onto a trap column (Acclaim1 PepMap 100 pre-column, 75 µm, 2 cm, C18, 3 mm, 100 Å, Thermo Scientific) and then separated on an analytical column (EASY-Spray column, 25 cm, 75 µm i.d., PepMap RSLC C18, 2 mm, 100 Å, Thermo Scientific) using solvent A, 0.1% formic acid in water and solvent B, 0.1% formic acid in ACN, at a flow rate of 300 nL/min and a column temperature of 45 °C. An estimated peptide amount of 1 µg was injected into the column, and the following gradient was used: 0 min 4% B; 3 min 4% B; 109 min 30% B; 124 min 45% B; 125 min 98% B; 130 min 98% B. The TMT node was utilized as follows: full MS scans at *m*/*z* 350–1400 with a resolution of 120,000 at *m*/*z* 200, a target AGC value of 3 × 10^6^ and IT of 50 ms, DDA selection of the 20 most-intense ions for fragmentation in HCD collision cell with an NCE of 34 and MS/MS spectra acquisition in the Orbitrap analyzer at a resolution of 45,000 (at *m*/*z* 200) with a maximum IT of 86 ms, fixed first mass of 110 *m*/*z*, isolation window of 0.7 Da and dynamic exclusion of 30 s.

#### 2.3.2. Spectral Library

Peptides were dissolved in 2% ACN, 0.1% TFA, and spiked with iRT peptides (Biognosys AG, Zurich, Switzerland) in a 1:10 dilution (iRT:peptides). First, a spectral library using DDA was built using the same LC-MS/MS system as the global proteome analysis with the same type of trap and analytical column, flow rate, temperature, and solvents. The gradient used was the following: 0 min 4% B; 7 min 4% B; 139 min 30% B; 154 min 45% B; 155 min 98% B; 160 min 98% B. The MS parameters were set as follows: selection of the 15 most intense ions for fragmentation, full MS scans at *m*/*z* 375–1750 with a resolution of 120,000 at *m*/*z* 200, a target AGC value of 3 × 10^6^ and IT of 100 ms, fragmentation in HCD collision cell with normalized collision energy (NCE) of 25 and MS/MS spectra acquisition in the Orbitrap analyzer at a resolution of 60,000 (at *m*/*z* 200) with a maximum IT of 120 ms and dynamic exclusion of 30 s.

#### 2.3.3. Phospho-DIA

For DIA-MS, the phosphopeptides were separated using the same gradient and MS system as for the DDA analysis, and the iRT mix was added to the individual samples. The full scans were processed in the Orbitrap analyzer with a resolution of 120,000 at (200 *m*/*z*), an injection time of 50 ms, and a target AGC value of 3 × 10^6^ in a range of 350 to 1410 *m*/*z*. Fragmentation was set to 54 variable isolation windows based on the density distribution of *m*/*z* precursors in the previously built spectral library. MS2 scans were acquired with a resolution of 30,000 at 200 *m*/*z*, an NCE of 25, a target AGC value of 1 × 10^6,^ and 200 *m*/*z* as fixed first mass.

### 2.4. Proteomic Data Processing

#### TMT 11 Plex Quantification of Proteomic Data

The global proteomic experiment generated a total of 375 raw files that were processed with Proteome Discoverer 2.3 (Thermo Fisher Scientific) using the Sequest HT search engine. The search was performed against the Homo sapiens UniProt revised database (downloaded 2018-10-01) with isoforms. Cysteine carbamidomethylation (+57.0215 Da) and TMT6 plex (+229.1629 Da) at peptide N-terminus and lysine were set as fixed modifications while methionine oxidation (+15.9949 Da) and N-terminal acetylation (+42.0105 Da) were set as variable modifications; peptide mass tolerance for the precursor ions and MS/MS spectra were 10 ppm and 0.02 Da, respectively. A maximum of two missed cleavage sites were accepted and a maximum false discovery rate (FDR) of 1% was used for the identification of PSM, peptide, and protein levels using all samples of the proteomic dataset. The Proteome Discoverer software v. 2.3 allowed the introduction of reporter ion interferences for each batch of TMT11 plex reagents as isotope correction factors in the quantification method. The peptides that could be uniquely mapped to a protein were used for relative protein abundance calculations.

These search results were imported into the Perseus software v. 1.6.6.0. To correct for experimental differences related to sample handling and other biases, such as column changes, the protein intensities were log2 transformed and centered around zero by subtracting the median intensity in each sample. To allow for the comparison of relative protein abundances between the different batches of TMT11 plex, the protein intensities from the pooled references sample (in channel 126 in each batch) were subtracted from each channel in the corresponding batch to obtain the final relative protein abundance values.

### 2.5. Phosphoproteomic Data Processing

#### DIA Phosphopeptide Quantification

The phosphoproteomic spectral library was generated from 45 DDA raw files in the Spectronaut X platform (Biognosys AG) [24] against the Homo sapiens database from Uniprot (downloaded 2019-01-15) with isoforms. The following parameters were used: cysteine carbamidomethylation (+57.0215 Da) as fixed modification and methionine oxidation (+15.9949 Da), N-terminal acetylation (+42.0105 Da) and phosphorylation (+79.9663 Da) on serine, threonine, and tyrosine were selected as variable modifications. A maximum of two missed cleavages were accepted. Precursor mass tolerance was set to 10 ppm and for the MS/MS fragments, it was set to 0.02 Da. Between 3 and 25 fragments were collected per peptide. The phosphosite localization algorithm was set so that phosphosites with a score equal to or higher than 0.75 (i.e., 75% accuracy of determination of the phosphate localization) were considered as Class I. Filtering was performed at 1% FDR at PSM, peptide and protein levels for the whole phosphoproteomic dataset.

The 122 DIA raw files were analyzed in Spectronaut X. In the transition settings, charges +2 and +3 were set for the precursor ions, and +1, +2, and +3 were set for the b- and y-ion products, with a mass tolerance of 0.02 Da. Both the precursor and protein q-value cutoffs were set to 0.01, and the peptides were quantified based on the intensity of the MS1 signal precursor. In all samples, the retention time alignment was performed with spiked-in iRT peptides (Biognosys, Schlieren, Switzerland). From the database search, a total of 45,356 phosphosites in 29,484 phosphopeptides were identified, with an average of 18,722 phosphosites identified per sample. The data displayed 58.7% missing values in the phosphosite abundance. The data were exported into the Perseus software v. 1.6.2.3 [25]. Valid value filtering was applied and all phosphosites with more than 5% missing values were removed. The data were then log2-transformed and centered around zero by subtracting the median intensity in each sample. For those phosphosites with less than 5% missing values, the phosphosite abundance values were imputed by applying the K-nearest neighbor method, resulting in 4644 phosphosites in each patient used for the kinase analysis, ICA, and survival analyses.

### 2.6. Immunohistochemistry

For the immunohistochemical study, 42 primary melanoma tissues were used. The fixation of the tumor material was performed after the surgical removal of the melanoma tissue with 4% cc. buffered formaldehyde (volume ratio 1 tissue/10 fixative). Samples were then dehydrated in xylene/ethanol solution, embedded in paraffin, and stored at room temperature. Sections of 10 μm were used for further immunohistochemical analysis. Representative tissue areas from paraffin-embedded blocks were selected based on the HE-stained slides, then 5 mm circumferential columns were put into the tissue microarrays (TMAs) in an ordered manner. From TMAs, 3.5 µm sections were placed into an automated immunostainer (Leica Bond Max, Leica Biosystems, Nussloch, Germany) using standardized deparaffinization, rehydration, and staining processes based on the automatized “Polymer No Enhancer–Bond Polymer Refine IHC protocol no enhancer” protocol in Leica Bond Max (Leica Biosystems instrument).

Antibodies against ADAM10, CDK4, CTNND1, DDX11, FGA, HMOX1, SCAI, PAEP, and PIK3CS3 were applied in dilution series (Appendix A). For visualization, a high-affinity polymer-based, AF-linked secondary antibody was used with a Fast Red chromogenic substrate. For negative controls, open containers were filled with primary antibody diluent without primary antibody. Before coverslipping, slides were counterstained with hematoxylin. All IHC stages were automatized.

The colorimetric immunostained slides were scanned by a 3D Histech slide scanner (Pannoramic MIDI, 2010 3D Histech Ltd., Budapest, Hungary). The digitized images served as a basis for the densitometry quantification of the antibody expression using the Image Pro Plus software (version 6.0). Multicolor pictures were converted into a grayscale spectrum. Then, representative areas of the cell cytoplasm and/or nucleus of melanoma and stromal cells were measured separately based on the strongest antibody expression in the grayscale spectrum. The continuous scale variables were collected for statistical analysis.

## 3. Results

### 3.1. Proteogenomic Map and Classification of Treatment-Naïve Melanoma Metastases

Global proteomic and phosphoproteomic analyses were performed on 142 metastatic melanoma samples from lymph nodes (126), cutaneous (1), subcutaneous (7), visceral (3), and of uncharacterized (5) origin (Appendix A). The analysis was supplemented with clinical data and a histopathological assessment of nearby cancer tissue (Appendix A). In addition, the proteomic and phosphoproteomic data were integrated with a previously published transcriptomic dataset from matched tumors [26]. A total of 12,695 proteins and 45,356 phosphosites were quantified, with 8124 proteins and 4644 phosphosites quantified in every sample (Appendix A). Several metrics were analyzed to assess the reliability of the proteomic workflow and the quality of the generated data (Appendix A).

Two independent studies have classified melanoma tumors based on transcript levels [7,27]. Since proteins are the functional entities in the cell, we investigated whether proteomic data could improve the classification of melanoma. Using consensus clustering (Appendix A), we identified five major melanoma subtypes, which were classified as extracellular (EC, n = 23), extracellular-immune (EC-Im, n = 26), mitochondrial (Mit, n = 30), mitochondrial-immune (Mit-Im, n = 23), and extracellular-mitochondrial (EC-Mit, n = 16) according to their characteristic enrichment in Gene Ontology (GO) terms and Kyoto Encyclopedia of Genes and Genomes (KEGG) pathways (Figure 1A and Appendix A).

We found significant associations between the proteomic and the two transcriptomic subtypings [7,27], displaying a mutual validation of the three classifiers (Figure 1D). Metastases belonging to the immune transcriptomic subtypes were divided into the EC-Im and Mit-Im proteomic subtypes. In addition, the MITF-low and proliferative subtypes were separated into the EC and the EC-Mit subtypes, and tumors within the Pigmentation were mainly divided into the two Mit subtypes. In contrast, the Keratin transcriptomic subtype was mainly split among the Mit and EC subtypes.

Pathway enrichment analysis of the proteomic subtypes at protein and phosphoprotein levels showed upregulation of oxidative phosphorylation, ribosome, metabolic, and RNA-related pathways in the Mit, Mit-Im, and EC-Mit subtypes (Figure 1A). Pathways associated with extracellular matrix organization, complement, and coagulation pathways were enriched in EC, EC-Im, and EC-Mit subtypes, suggesting a more invasive phenotype [28]. The EC-Im and Mit-Im subtypes displayed enrichment of immune signaling with upregulation of antigen processing and presentation and T cell receptor signaling pathways, including higher expression of PDL1 relative to Mit and EC-Mit (Appendix A). At the transcript level, the EC-Mit subtype showed pathways of increased RNA activity, while in both immune subtypes, the antigen processing and presentation pathway was enriched.

To identify subtype signatures, we used Independent Component Analysis (ICA) to select the top ten most contributing proteins to the independent components (ICs) significantly correlated with the proteomic subtypes (Appendix A). In the EC-Im subtype, we identified upregulation of the antigen receptor-mediated signaling pathway, emphasizing the immune system’s involvement in this subtype. Furthermore, positive cell–cell and cell–matrix adhesion regulation was seen, contributing to the extracellular matrix component of this subtype. Several proteins from the S100 family (S100P, S100A12, S100A8, and S1009) were identified in the EC subtype, suggesting their potential as markers for this subtype. In addition, enrichment of neutrophil degranulation was observed, which may contribute to poor prognosis and short overall survival in melanoma by upregulation of proliferation, angiogenesis, and matrix remodeling [29]. The EC-Mit signature included a set of downregulated proteins, but no pathway enrichment was detected, not even when all the proteins (8572) of this IC were submitted for enrichment analysis. In the protein signature associated with the Mit subtype, we found enrichment in essential mitochondrial proteins, such as OAT [30], VDAC2 [31], and SHMT2 [32]. High expression of S100A1 was apparent, pointing to a unique and strong association between this melanoma marker and the Mit subtype. The top ten signature proteins in the Mit-Im subtype were associated with a downregulation of the complement and coagulation cascade rather than enrichment in a mitochondria-related signature or an upregulation of protective immune mechanisms. When upregulated in the tumor microenvironment, the complement and coagulation cascade may enhance tumor growth and increase metastasis, and it is suggested to contribute to epithelial-mesenchymal transition (EMT) [33,34]. In addition, we could see an upregulation of these proteins in the EC subtype, again highlighting the molecular differences between these two subtypes.

### 3.2. Proteomic Subtypes Associate with Clinical and Histological Features

We found differences in histopathological features among the proteomic subtypes. (Figure 1B,C). The Mit-Im subtype was associated with higher lymphocyte density (Fisher exact test, FDR = 0.0003) than the non-immune subtypes. Both immune subtypes displayed a higher lymphatic score (sum of lymphocyte distribution and density) than the rest of the metastases (ANOVA test, FDR < 0.01). Tumor cell content was higher in samples belonging to Mit-Im, Mit, or EC-Mit when compared to samples within the EC-Im and EC subtypes (Kruskal–Wallis test, FDR < 0.005). Adjacent lymph node and necrosis content were higher in the EC-Im and EC subtypes compared to the others (Kruskal–Wallis test, FDR < 0.05). The EC, in particular, where 70% of the samples with the highest necrosis content (>20%) were found. In contrast, connective tissue content was higher in the EC than in the EC-Mit subtype (Kruskal–Wallis test, FDR = 0.046).

The histological images of the Mit subtype metastases showed solid nests of highly aggressive tumor cells with broader eosinophilic cytoplasms and atypical nuclear features (Figure 1B). This phenotype reflects the picture often seen in more differentiated tumors. In the EC subtype, we observed an abundance of the intercellular matrix as desmoplastic stromal change with a pauci-cell matrix, typical characteristics of phenotype switching in melanoma. Mit-Im and EC-Im appear to be subclasses of the highly aggressive Mit and EC subtypes, respectively, where the presence of adaptive immune cells may indicate a more favorable clinical behavior, pointing towards a possible antitumor effect of adjacent lymphatic tissue [35]. Histologically, the EC-Mit group also represents a dedifferentiated state, which may be an intermediate state between the more differentiated Mit group and dedifferentiated EC groups. This group shares features of epithelioid and stromal-rich areas within the tumor.

We found an association between the patients in the group consisting of the EC, EC-Mit, and Mit subtypes and disease stage IV (Fisher exact test, FDR = 0.012, Appendix A). Patients with tumors in the EC, EC-Mit, and Mit subtypes had an increased risk of developing distant metastases and shorter survival times from the detection of the first metastasis. They were associated with overall survival (OS) of less than five years when compared to patients with metastases belonging to the Mit-Im and EC-Im subtypes (Fisher exact test, FDR < 0.03). Other significant associations between the subtypes and histopathological and clinical parameters can be found in Appendix A. In addition, as expected, Kaplan–Meier analysis showed that patients within the immune subtypes (Mit-Im + EC-Im) had a significantly better prognosis than patients with metastases belonging to other subtypes (EC + EC-Mit + Mit) (Figure 1E). A similar trend was observed in the survival analysis of these two groups when considering patients in stage IIIB and IIIC, respectively (Appendix A).

Interestingly, there were no significant associations between the proteomic subtypes and the presence of BRAF V600E or NRAS Q61K/R mutations, which indicates that the subtypes are not driven by the main mutational events occurring in melanoma, instead providing an orthogonal classification. Also, several metastases from patients in disease stage IIIA/IIIB to IIIB clustered with metastases from patients in stage IV across the more aggressive subtypes EC, EC-Mit, and Mit (Figure 1A). Thus, the proteomic subtypes classify melanoma lymph node metastases beyond the level of clinical staging and genomic driver mutations.

### 3.3. Expression of Melanoma and Phenotype Switching Markers Across the Subtypes

Protein expression levels of known melanoma markers [36], such as MITF, MLANA, PMEL, TYR, and SOX10, had the lowest expression in EC-Mit samples (ANOVA, FDR < 6 × 10^−4^) and had a significantly lower expression in the EC-like subtypes when compared to the Mit-like subtypes (Figure 2A). MITF is one of the key regulators of melanoma differentiation and has been implicated in playing a role in dedifferentiation, commonly referred to as melanoma phenotype switching. MITF expression has been associated with survival, cell cycle control, invasion, senescence, and DNA damage repair [37,38]. Higher expression of MLANA and TYR have also been linked to phenotype switching in melanoma and is associated with a more differentiated phenotype [39,40,41]. The transcript and protein levels of these genes were higher in the Mit and the Mit-Im than in the other subtypes (Figure 2A,B).

The protein expression of the EMT hallmark genes, CDH1 (E-Cadherin) and CDH2 (N-Cadherin), were analyzed to evaluate the phenotype-switching traits observed at the histological level. We found differential expression of these proteins among the subtypes resulting in a significantly higher relative abundance relationship of CDH2 and CDH1 (CDH2/CDH1 ratio) in the EC, EC-Im, and EC-Mit subtypes (ANOVA, adjusted *p*-value < 0.05) compared to the Mit and Mit-Im metastases, which indicated a switched gene expression from CDH1 to CDH2 in these subtypes (Figure 2C and Appendix A). The protein and transcript levels of known markers of melanoma phenotype-switching [40] were also analyzed. The gene expression of FAP, ERBB3, FOSL1, and SERPINE1, which have been associated with a dedifferentiated state, were significantly higher in the EC or EC-Im compared with the Mit and Mit-Im subtypes. Also, protein levels of SOX10 and GPR143, associated with a more differentiated state, were significantly higher in the Mit and Mit-Im compared to the EC or EC-Im subtypes. Moreover, there was a significant differential expression of the SOX4, FOSL1, and SERPINE1 markers between the EC and EC-Mit proteomic subtypes. A trend towards higher protein levels of AXL, EGFR, and NGFR and lower MITF/AXL protein ratios (ANOVA, adjusted *p*-value < 0.05) were found for the EC, EC-Im, and EC-Mit subtypes compared to the Mit and Mit-Im metastases (Figure 2A,D). This gene expression pattern has been linked to resistant tumors with a dedifferentiated and invasive phenotype [13,42,43]. Recent studies have targeted AXL and EGFR to improve sensitivity to BRAF V600-targeted treatment by interfering with phenotype switching and stabilizing melanocyte differentiation to enhance immunotherapy and targeted therapy outcomes [44,45,46,47]. As has been observed before in melanoma lymph node metastases, VIM did now show differential expression among the subtypes [48]. However, unlike what has been observed for primary tumors at the protein level and metastases at the transcript level, we found no significant difference for SNAI1/2 and ZEB1/2 transcription factors among the metastatic subtypes, which might reflect the peculiarities of phenotype switching in melanoma at the protein level. It might also be due to various phenotypes and differentiation states in heterogeneous metastases, as suggested before [40].

Overall, the analysis supports a more differentiated state for the Mit and Mit-Im subtypes and a dedifferentiated state for metastases of the EC, EC-Im, and EC-Mit subtypes, matching the histological observations. The presence of several dedifferentiated states is consistent with the increasing evidence that phenotypic transitions encompass more complex dynamics than the highly differentiated and dedifferentiated states [49].

The EC-Mit metastases showed significantly lower expression of the proliferation marker Ki67 compared to the other subtypes, suggesting a more invasive phenotype. Interestingly, there was no significant difference in proliferation among the EC and EC-Im subtypes compared to the more differentiated Mit and Mit-Im subtypes (Figure 2E), which challenges the assumed association between invasiveness and proliferation, pointing to a non-exclusive behavior of these features as highlighted in previous studies [50].

### 3.4. Selective and Coordinated Regulation of S100 Family Members

S100 antigenicity is used to identify poorly differentiated metastatic melanoma, and this protein family has been implicated in cell proliferation, metastasis, angiogenesis, invasion, and inflammation [36,51,52,53,54,55,56]. Selected members of the S100 protein family were divided into two groups (Figure 2F,G). The first group, consisting of S100P, S100A8, S100A9, and S100A12, was significantly upregulated in the EC subtypes (FDR < 0.003). S100P and S10012 have a diagnostic and prognostic value in many human cancers [54,57], while the contribution of S100A8 and S100A9 to melanoma biology is not fully understood. In the second group, S100B, S100A1, and S100A13 were significantly upregulated in both mitochondrial subtypes compared to the EC subtypes, with FDR = 0.0085 and *p*-value = 0.031, respectively. S100B inhibits p53 phosphorylation and is a serum biomarker of advanced disease stage, poor therapeutic response, and low patient survival [54,58,59]. S100A1 is known for dysregulating proliferation [55] and interacting with the mitochondria [60]. Overall, the results suggest a selective and coordinated regulation for certain S100 family members within the different proteomic subtypes, which, together with other melanoma markers, may be used to discriminate between proteomic subtypes with poor prognosis.

### 3.5. Expression of Targetable Phosphorylated Kinases Among the Subtypes

To identify potential therapeutic targets specific to each proteomic subtype, the phosphoproteomic data on kinases were used as “potential kinase activation surrogates” [61,62,63,64]. We found five phosphosites significantly upregulated in kinases in the EC-like subtypes: AAK1, MARK2, ROCK1, BCKDK, and RIPK3 (ANOVA *p*-value < 0.05) (Figure 2H). Phosphorylation at Thr208 of MARK2 was located in the activation loop, which may lead to loss of cell polarity through interfering with microtubule stability and to an unfavorable prognosis, which has been observed in other cancers [65] (Appendix A). In the EC-Mit, Mit, and Mit-Im subtypes, we found upregulation of phosphosites in the kinases BRD3, RAF1, SRPK1, GSK3A, and CMKK2. In GSK3A, the phosphorylated Ser278 and Ser282 are located in the activation loop, flanking the known activation site Tyr279 (Appendix A). GSK3A was recently linked to cancer stem cells and drug resistance [66]. Decreasing GSK3A protein levels using siRNA and pharmacological targeting reduced melanoma tumor development in murine models [67]. Thus, GSK3A could be a promising target for subtype-specific treatment in melanoma.

### 3.6. Proteomic Signatures and Patient Survival Stratify BRAF V600 Positive Metastases

A previous study of patients with metastatic melanoma quantified the level of the BRAF V600E-mutated protein [68]. Higher expression was associated with lower immune response and increased cell proliferation, leading to tumor progression and shorter survival. This uncovered heterogeneities in tumor morphology, protein profiles, and clinical outcomes in a limited subset of BRAF V600-positive melanomas. Here, we investigated whether these results could be extrapolated to the 49 BRAF V600-mutated metastases of this study, of which 67% lacked data on BRAF mutation expression (Appendix A). InGRiD (Integrative Genomics Robust iDentification of cancer subgroups, see experimental) was used to classify patients into low-, medium-, and high-mortality-risk groups. Patients with different mortality risks were identified and classified into low-, medium-, and high mortality-risk groups (Figure 3A). Notably, eight out of nine patients with high BRAF V600E-mutated protein levels were in the high- and medium-risk groups (Figure 3B). In contrast, most patients with low levels of the BRAF V600E mutation had a low mortality risk.

The proteins involved in classifying these risk groups were mainly enriched in pathways related to metabolism, the immune system, and signal transduction (Figure 3C, Appendix A). Neutrophil degranulation, the complement cascade, mRNA transcription, TGF-beta signaling, and DNA repair were positively related to a high mortality risk. On the other hand, signal transduction, vesicle-mediate transport, mitochondrial fatty acid beta-oxidation, fatty acid metabolism, and nucleotide metabolism were positively related to a low mortality risk. We also investigated whether disparities between BRAF patients with low- and medium-high mortality risks were apparent among the histological features. There were no significant differences in tumor cell content between the low- and medium-high mortality-risk groups (Figure 3D). However, low-risk metastases, as opposed to the medium-high-risk group, were characterized by higher lymphatic scores (ANOVA test, FDR < 0.05) and adjacent lymph node content (Kruskal–Wallis test, FDR < 0.05) as well as lower tumor-derived connective tissue and necrosis content (Kruskal–Wallis test, FDR < 0.05).

In a previous study, we observed a tendency towards larger cells with multiple nuclei in the metastases of the low BRAF V600 group. This feature has been associated with cellular senescence [69,70,71]. Transcriptomic studies with cultured melanoma cells and mouse models have found a link between oncogene-induced senescence (OIS) and an increase in the MHC II molecules, which is associated with a favorable disease outcome [72,73]. In the low-mortality-risk group, in agreement with the higher lymphatic scores assigned to these tumors, we found an upregulation of most proteins related to antigen processing and presentation by MHC II (Figure 3E). This proposes an OIS-like phenotype that could act as a tumor suppressor and lead to a better outcome for this group.

Enrichment of the differentially expressed proteins and phosphosites between the two mortality risk groups showed an upregulation of the interferon-gamma response in the low-mortality-risk group (Appendix A), which may also induce senescence and activate MHC I antigen presentation [74,75]. Indeed, we found an upregulation of many proteins within the MHC I pathway in the low-mortality-risk group (Figure 3E). This suggests an increased susceptibility to cell-mediated cytotoxicity in this group and a possible contribution to the OIS-like phenotype, resulting in a better prognosis [76].

Several proteins from the KEGG cellular senescence pathway, such NFKB1, MAP2K1, RASSSF5, MRE11A, and RAD50, were upregulated in the low-mortality-risk group, whereas CDK4, which, when bound to Cyclin D inhibits Rb, was downregulated (Figure 3F). Moreover, a significantly increased phosphorylation of CDKN1B at Ser-10 was observed (Figure 3F). This is the most important phosphorylation site in resting cells as it inhibits CDK2 activity, prohibiting G1 to S phase cell cycle transition.

To further support the OIS-like state in the low-mortality-risk group, JmjC demethylases, such as KDM4A and 5B, known to regulate senescence directly or indirectly, were found to be downregulated in these tumors (Figure 3F) [77]. KDM5B has been shown to endorse the demethylation of H3K4me3/2, resulting in the silencing of E2F target gene promoters through direct interaction with Rb [78,79]. The demethylase activity of KDM5B is crucial for DNA damage repair and genomic instability, whereby the inhibition of KDM5B leads to the activation of p53, inhibiting cell proliferation [80,81,82]. The downregulation of KDM4A and KDM5B is known to activate the p53 pathway, triggering senescence and knockdown of KDM4A that can lead to the buildup of promyelocytic leukemia (PML) nuclear bodies, a marker of senescence, also upregulated in the low-mortality-risk group (Figure 3F) [83]. Regulation of the PML protein, a known tumor suppressor, largely depends on several phosphorylations of the protein. We found an upregulation of phosphorylation at Ser-8 and Ser-36, which increases PLM protein accumulation, and at Ser-403 and Ser-518, which promotes PML protein degradation in the BRAF low-mortality-risk group, suggesting an increased protein turnover (Appendix A) [84,85,86,87]. KDM4A and KDM5B can be considered proto-oncogenes, and targeting these demethylases could potentially result in tumor suppression, which makes them attractive therapeutic targets in melanoma.

A significant association was found between the low-risk mortality BRAF subgroup and the proteomic subtypes with better prognosis (EC-Im and Mit-Im) (Fisher exact test, FDR = 0.002). In comparison, the medium- and high-risk BRAF subgroups were significantly associated (Fisher exact test FDR = 0.002) with the subtypes having worse prognoses (EC, Mit, and EC-Mit) (Figure 3G). The medium-high BRAF V600 mortality-risk group was also associated with an increased risk of developing distant metastases, shorter survival from the detection of the first metastasis, and shorter overall survival (Appendix A).

The differences in protein profiles, biological pathway enrichments, and histological features support a heterogeneous BRAF mutation expression. This is defined by two major groups of BRAF V600 positive metastases linked to the escape from or exposure to immune surveillance. For the latter, we propose an OIS-like phenotype as a tumor-suppressing mechanism in patients with lower mortality risks, contributing to their better prognosis.

### 3.7. Melanoma-Associated Single Amino Acid Variants (SAAVs)

Melanoma is considered the cancer with the highest mutational burden [7,88]. Interestingly, the study of protein expression of genomic alterations in melanoma tumors has been largely overlooked. By further exploring the melanoma proteome, we identified 1015 SAAVs in 828 proteins in the lymph node metastases (Appendix A). Interrogation of the CanProVar database and the Cancer Gene Census resulted in 27 and 24 SAAVs related to cancer or belonging to genes with mutations implicated in cancer, respectively (Appendix A). In addition, 30 SAAVs were predicted to be cancer-promoting by FATHMM [89].

We matched identified proteins with SAAVs to signaling pathways and biological processes recurrently dysregulated in melanoma (Appendix A) [90]. The highest number of SAAVs was found in the PI3K/AKT and MAPK signaling pathways. Besides the well-known role of mutations in members of the MAPK pathways, such as BRAF and NRAS, an increasing number of studies have linked gene polymorphism and genetic variants to the members of the PI3K/AKT signaling pathway as susceptibility for cancer development [91,92,93]. Most of the SAAV-affecting proteins of the PI3K/AKT signaling pathway were structurally or functionally associated with the ECM. Indeed, Reactome and KEGG pathway analyses of proteins with SAAVs showed enrichment in three general categories, with ECM-related processes as the most enriched pathway. This was followed by cellular metabolism and the complement and coagulation cascade (Appendix A). Less than 2% (19) of all the SAAVs originated from samples with <50% tumor content, indicating that the melanoma cells likely produce the ECM-related protein variants. This finding aligns with a recent Pan-Cancer genomic study, which revealed that a higher copy number and more missense mutational alterations are present in the ECM genes compared with the rest of the genome [94].

Quantitative analysis of the SAAVs expression resulted in 52 differentially expressed variants across the proteomic subtypes (ANOVA, FDR < 0.05), where the highest number of overexpressed SAAVs was found in the EC subtype (Figure 4A). These SAAVs underlined an active role in the remodeling of the ECM and dysregulation of the complement and coagulation pathway, processes that are likely associated with phenotype switching of melanoma cells. This is in agreement with the enriched pathway results for all identified variants.

Appendix A shows the variants and wt counterparts (canonical sequences) in the analyzed metastases, considering the total (top panel) and the relative (middle panel) PSM number of the corresponding peptides as surrogates of protein expression. As expected, wt PSMs outnumbered those of the SAAVs, which indicated a higher abundance of the wt proteins. However, this trend had several exceptions, including wt peptides not being detected or surpassed by SAAV PSMs. Matched SAAVs with corresponding variant allele frequency (VAf) in the European population revealed a fraction of SAAVs (67, 6.5%) dominating in PSMs but with low (<0.3) or not reported VAfs (Appendix A bottom panel). It was also found that 34% of identified SAAVs have VAf > 0.5. These results indicated (1) the need to further filter the list of identified SAAVs to uncover those more associated with melanoma and (2) that many of the identified variants are common or are the major proteoforms among the population; however, their identification is usually overlooked in proteomic studies due to the use of protein databases with the canonical sequences.

To predict which SAAVs are significantly related to melanoma, we searched the dbSNP Short Genetic Variations database for the European population’s corresponding allele frequencies (VAf). As a result, we generated a signature of 167 melanoma-associated SAAVs classified into levels 1, 2, and 3, according to genomic data and proteomic evidence (Figure 4B,C, Appendix A). These SAAVs displayed the same pathway enrichment as described above (Figure 4D). The signature captured 26 critical SAAVs among the genes and mutations reported by CanProVar and The Cancer Gene Census or predicted by FATHMM (Figure 4E). This included the melanoma driver mutations NRAS Q61K/R, BRAF V600E, CDKN2A P114L, and HRAS G13 (Figure 4C). These findings served as evidence to support the proposed signature of melanoma-associated SAAVs. Interestingly, we also found five SAAVs in heavy chains of muscle myosin II complex (MYH1 E1853K, MYH2 E486K, MYH4 N1627I, MYH8 E932K, and MYH13 D1765N), which presented the highest number of variants originated from the same loci 17p13.1 in our dataset (Appendix A). This is the same loci of TP53, a frequently mutated tumor suppressor credited as a hallmark of cancer. Myosin II is required for cell contractility, cytoskeleton reorganization, and cytokinesis. Different reports associated mutations and polymorphism in heavy chains of muscle myosin II with cancer predisposition [95] and non-cancer-related diseases [96,97]. We hypothesize that these variants are reactivated by melanoma cells as part of cytoskeletal remodeling by myosins [98].

Tumor mutational burden can predict response to immunotherapy in melanoma [99], which suggests that mutated peptides binding to MHC I class molecules can be the primary origin of target antigens. To investigate neoantigens with the potential to express identified SAAVs, we aligned the amino acid sequence of the SAAV-bearing peptides to a large experimental dataset of a melanoma-associated immunopeptidome [100]. This resulted in 56 SAAVs that could be presented as variant peptide ligands by the HLA I complex (Appendix A). According to the NetMHC prediction tool [101], 61 SAAV-altered peptides ranked better as HLA class I neoepitopes than their wild-type counterparts (Appendix A). This included 15 cases where the latter fell out of the specified threshold (%Rank_EL < 2) for binding prediction (Figure 4F and Appendix A). The variant peptide ligands CYB5R1 N44S, CD300LF Q218R, GCA S80A, and QARS N285S were the best candidates for the HLA allotypes HLA-A26:01, HLA-B07:02, HLA-B15:01, and HLA-B58:01, respectively. Among the significant HLA I peptide ligand variants, we found CDKN2A P114L, ARL6IP6 R56L, GCA S80A, LOXL1 R141L, CFL1 L111F, HSPB1 I179N, and TGFBI E126K, which are a part of the melanoma-associated SAAV signature defined in this study. The results indicated that knowledge of variant expression, supported by melanoma-associated immunopeptidome and MHC I binding prediction tools, may lead to the discovery of neoantigen candidates as targets of anti-tumor immune responses.

### 3.8. The Tumor Microenvironment Is a Key Player in Disease Progression

Differences in the content of adjacent lymph nodes and the tumor-derived connective tissue between the BRAF V600 mortality-risk groups and among the proteomic subtypes pointed to a link between the TME and patient outcome. Therefore, a subset of 29 samples with less than 50% tumor cells was analyzed. Univariate receiver operating characteristic (ROC) curve analysis separated groups of patients with differences in survival (DSS < or ≥3 years) (Figure 5A). The groups were named high or low lymph node (HLN or LLN) and high or low connective tissue (HCT or LCT) based on the content of these histological features in the tumors. With minor differences, the HLN and LCT or LLN and HCT groups allocated the same patient subsets (Figure 5B,C), reflecting the significant correlation (Spearman −0.84, *p*-value = 1.64 × 10^−8^) and complementarity between LN and CT contents in this subset.

Patients within the HLN and LCT groups had a better prognosis, while the opposite was found for patients in the LLN and HCT groups (Figure 5D). Strikingly, Cox regression models showed that both LN (Cox coefficient = −1.661, *p*-value = 0.001) and CT (Cox coefficient = 1.718, *p*-value = 0.001) contents were better indicators of prognosis than disease stage (Cox coefficient = 1.265, *p*-value = 0.008). In addition, multivariate Cox regression analysis considering age, gender, and disease stage showed an increased risk of developing distant metastasis for the LLN group compared to the HLN group, with a Hazard ratio (HR) of 5.96 (95% CI: 1.629 to 28.96, *p*-value = 0.021). A similar analysis based on OS showed that patients with metastases in the LLN group have shorter life spans than those in the HLN group (HR = 19.91, 95% CI: 3.350 to 173.4, *p*-value = 0.0023).

The TME of the HLN (also LCT) group was dominated by immune cells, both surrounding and infiltrating the tumor. On the contrary, the LLN (also HCT) group displayed large areas of stroma and fatty tissue mixed with tumor cells (Figure 5E). These observations were supported by cell-specific transcript signatures based on a single-cell study of melanoma TME [49,102], which were confirmed at the protein level in our study (Figure 5F). The HLN group was enriched in markers mainly expressed by B and T lymphocytes, macrophages, and endothelial cells (Figure 5F). The enriched pathways were related to antigen processing and presentation, ribosome, and B and T cell receptor signaling at both proteomic and transcriptomic levels (Figure 5G and Appendix A). On the contrary, the LLN group displayed enrichment in markers of cancer-associated fibroblasts (CAFs), macrophages, and endothelial cells. Markers of CAFs, such as collagens, complement components, and growth factor proteins, were highly upregulated. This group was also enriched in pathways related to the complement and coagulation cascades, EMT, ECM organization, and collagen chain trimerization and degradation. Dysregulated collagen synthesis and assembly are common driving factors in many cancers [103].

Among the upregulated proteins in the LLN group, LOX (lysyl oxidase enzymes), which catalyze the crosslinking of collagens and elastin, increasing the tissue stiffness, has been proven to promote tumor progression through increased integrin signaling and EMT [103]. In addition, upregulation in the fibroblast activation protein (FAP), which is known to be expressed in tumor mesenchymal and epithelial cells, enhances migration and invasion [104,105,106]. Consequently, the TME and its cell composition fundamentally influence the disease progression. This is in line with previous studies in breast and colon cancer, where the tumor-stroma ratio in lymph node tissue was linked to prognosis, and a high stromal score (>50%) was associated with a more aggressive tumor progression [107,108,109]. The results fuel the need to explore such variables in larger cohorts since the histopathological assessment could be readily implemented in clinical practice and used for better-informed medical decisions.

### 3.9. A Molecular and Pathway-Level Understanding of Melanoma Histopathological Features

To identify relationships between clinical and histopathological features and biological pathways across omics datasets, we used ICA combined with gene set enrichment analysis (GSEA) (Appendix A). Variables such as tumor cell content, adjacent lymph node tissue, and necrosis contents were significantly related to many Reactome pathways supported simultaneously by proteomic, phosphoproteomic, and transcriptomic data (Appendix A). Furthermore, the association of one or several features to particular ICs exhibited direct or reverse relationships towards one another (Figure 6A,B). For example, we found that proteomic IC 48 was positively associated with closely related parameters, such as adjacent lymph nodes, lymphocyte density, and lymphocytic scores. Still, it was also negatively associated with tumor content. Similar associations were observed in ICs 10 and 92 for the phosphoproteomic and transcriptomic datasets, respectively. These ICs shared enrichment in immune-related pathways, such as TCR signaling, PD-1 signaling, IFN signaling, neutrophil degranulation, and complement and coagulation cascades. In addition, specific pathways attributed to protein, transcript, and phosphoprotein expression were found, including apoptosis, antigen processing, and presentation and signaling by Rho GTPases, respectively. The latter has also been associated with triggering multiple immune functions [110].

### 3.10. Multi-Omics Data Pinpoint Proteins Related to Patient Survival

To find biomarkers related to patient survival, we performed two different analyses. First, outlier analysis (Appendix A), which is based on survival groups (0.5, 1, 3, >5 years), was utilized. For each of these survival groups, we selected the enriched transcript, protein and phosphosite outliers (Appendix A). Secondly, a regularized Cox regression (Appendix A) was performed to complement with additional markers of survival (Appendix A). As a result, we found 103 proteins, 44 transcripts, and 21 phosphosites significantly related to survival (outlier FDR < 0.05, Cox score > 30).

The proteins ADAM10, HMOX1, FGA, DDX11, SCAI, CTNND1, CDK4, PAEP, and PIK3CB were selected based on the above survival analysis, a comprehensive evaluation (Appendix A), and literature search (Appendix A). Three proteins (ADAM10, FGA, and HMOX1) were chosen based on specific phosphosites linked to survival. For FGA (Ser364) and HMOX1 (Ser229), the phosphorylation of these particular sites was related to poor survival, while ADAM10 phosphosite Thr719 was enriched in tumors from patients surviving longer. In addition, high CDK4, CTNND1, PAEP, DDX11, SCAI, and PIK3CB protein levels in the lymph node metastases were all related to poor prognosis.

To expand our understanding of these markers in earlier disease stages, we assessed their expression in an independent cohort of primary melanomas, which resulted in metastatic disease during patient follow-up. The tissue expression of the nine target proteins was evaluated by immunohistochemistry in a tissue microarray (TMA) platform generated from 42 patients (Appendix A) with primary treatment-naive melanomas progressing into either locoregional or visceral metastatic disease. The proteins were localized within the stroma/TME and the melanoma cells (Figure 7, Appendix A)

DDX11-N, SCAI-M, SCAI-S, ADAM10-M, ADAM10-S, and CDK4-S denote the IHC expression of proteins in specific cell types or cellular compartments. ‘S’ indicates expression in stromal cells, ‘M’ represents expression in melanoma cells, and ‘N’ represents nuclear protein expression. We found that ADAM10 and SCAI expression was increased in the primary melanomas from patients with OS > 5 years compared to tumors from patients with OS < 5 years (Mann-Whitney U test *p*-value < 0.05) (Figure 7C and Appendix A). A similar trend was observed in the stromal expression of CDK4 and tumoral DDX11 (*p*-value < 0.05, Figure 7C). In addition, spatial expression-based ROC curve analysis followed by Kaplan–Meier (K-M) analysis revealed that higher ADAM10, FGA, DDX11, PAEP, and SCAI expression in primary melanoma cells could be linked to more prolonged survival, as well as a higher stromal expression for ADAM10, CDK4, SCAI, HMOX1, and PAEP (Figure 7D and Appendix A).

To follow the dynamics of disease progression, we compared these proteins in up to nine pairs of matching primary melanomas and lymph node metastases (Appendix A). The expression of most proteins tended to increase from the primary tumor to the corresponding metastasis, except for cellular CTNND1 and both cellular and stromal ADAM10 (Figure 7E). In the metastases, CDK4 and DDX11 expression in melanoma cells were significantly higher (Wilcoxon-test *p*-value < 0.05), and HMOX1 and PAEP expression significantly increased (Wilcoxon-test *p*-value < 0.05 in both melanoma cells and the stroma/TME), while SCAI expression was significantly increased in the metastatic stroma (Figure 7E).

## 4. Discussion

Melanoma of the skin is heterogeneous regarding cancer cell behavior and the surrounding tumor microenvironment, including the lymphoid, histiocytic, and fibroblastic elements. Cells from the primary melanoma may disseminate early, undetectable, during a latent clinical phase [111]. Due to the considerable heterogeneity of melanoma tumors, patient stratification beyond the level of mutations is of utmost importance for improved clinical decisions [112]. The melanoma samples in the present study were collected prior to the era of immune checkpoint and targeted therapies. Consequently, the molecular and histological characterization of this treatment-naïve cohort makes it possible to study the biological characteristics of melanoma metastases in close connection with patient outcomes.

Here, we propose proteomic subtypes that integrate microenvironment players, such as the immune and stroma components, unlike transcriptomic studies on tumors or cell lines [7,12,13,96]. This holistic approach enabled refined subtyping of melanoma and significant associations with clinical and histopathological features. For instance, the EC-Im and Mit-Im subtypes were associated with slower metastatic spread and linked to a better prognosis. In contrast, the EC, EC-Mit, and Mit subtypes appeared to escape immune control and were accompanied by a more aggressive disease. Our results suggest that the subtypes can be distinguished by MS-based abundance of routinely used histopathologic protein markers, such as MITF, PMEL (HMB45), MLANA, and certain S100 protein family members [113]. The S100 proteins appear to have a selective and coordinated regulation across subtypes and are connected with melanoma phenotype switching, also observed across the subtype signatures obtained by ICA. Altogether, following rigorous validation, the histopathologic markers and the ICA-derived protein signature may serve as potential diagnostic biomarkers for patient stratification.

The proteomic subtypes unraveled the presence of different states of phenotype-switched melanoma cells, which is in line with prior observations regarding the plasticity and phenotypic diversity of melanoma, contributing to increased knowledge of the pathomorphology within the tumoral tissue [40,49,114]. Here, dedifferentiation appears to be linked to accelerated disease progression due to the association between the EC-Mit subtype and earlier development of distant metastases. Further, it is becoming increasingly clear that dedifferentiation is a mechanism of resistance to both immunotherapy and targeted therapy [115,116,117,118], and melanoma patients would primarily likely benefit from subtype classifications and selection of therapeutic strategy in accordance with subtype-specific features, especially for the more dedifferentiated EC, EC-Im, and EC-Mit subtypes.

In summary, the proteomic subtypes capture layers of complexity beyond the transcript level by association with molecular, functional, histopathological, and clinical aspects of the disease.

Based on activating specific metabolism-, immune system- and signal transduction-related pathways, patients bearing tumors with a BRAF V600 mutation could be stratified into subgroups with differing mortality risks. Our results point towards a link between low BRAF V600 expression and a better immune response mediated by a BRAF V600-induced senescence-like phenotype, where the OIS acts as a tumor suppressor through the upregulation of antigen presentation and interferon-gamma signaling, thus improving patient outcomes. The BRAFV600E protein expression appears to have prognostic significance independent of the therapy. For metastatic spread, the surrounding niche of TME is necessary for progression [106]. In a subset of patients with lower tumor cell content, we found a significant association of adjacent lymph node tissue and connective tissue with patient survival, both appearing to be independent prognostic markers in melanoma lymph node metastases. Assessment of tumor-derived connective tissue/tumor or adjacent lymph node tissue/tumor ratio can be accomplished in routine pathology on hematoxylin and eosin-stained slides. It might be helpful as a selection criterion for therapy, as suggested for colorectal cancer and breast cancer [100,101,102]. Thus, our study underscores the importance of further investigating these phenomena in larger cohorts to obtain deeper insights into the relationship between the tumor and its microenvironment and its association with clinical outcomes.

The classifiers mentioned above may pose challenges and opportunities from diagnostic and therapeutic perspectives, such as matching subtypes and TME subgroups with responses to immune checkpoint treatments or determining whether all patients harboring the BRAF V600 mutation benefit from targeted therapy. Moreover, these classifiers highlight the importance of the stroma tumor-infiltrating lymphocytes (sTILs) and their immune signature for survival.

Despite having the highest mutational burden among cancers, melanoma has a protein-level expression of mutations that remains highly underexplored. This study provides a comprehensive landscape of SAAVs, emphasizing their roles in dysregulated pathways and less-characterized variants. Notably, we found significant enrichment of SAAVs in ECM-related genes, implicating them in extracellular matrix (ECM) remodeling. This remodeling likely contributes to tumor microenvironment (TME) heterogeneity and disease progression, particularly in the EC subtype. We also identified over- and under-represented SAAVs in melanoma, with the TP53 locus (17p13.1) emerging as a hotspot for genomic instability, independent of TP53 mutations as has been found in various cancers, including melanoma [119,120,121]. Most identified variants at this locus belong to myosin II complex proteins, typically associated with skeletal muscle function but appear repurposed in melanoma for cytoskeletal remodeling. This likely drives increased cell migration, invasion, and metastasis, with motor proteins like myosins facilitating tumor progression, drug resistance, and immune evasion by modulating intracellular trafficking, cytokine secretion, and adaptation to hostile environments [98,122,123].

Furthermore, several variants rank higher as neoepitopes in the NetMHC prediction tool compared to wild-type peptide ligands, suggesting enhanced T-cell recognition within the TME. Among these, the CDKN2A P114L mutation stands out as a compelling neoantigen, with evidence linking it to improved immunotherapy responses, underscoring its potential as an immunogenic target [124]. These findings support the concept that strong neoantigens improve immune checkpoint inhibitor responses.

Decoding the functional roles of these SAAVs is critical for understanding their impact on protein function, melanoma progression, and patient survival. Experimental validation through peptide-MHC binding assays, immunopeptidomics, and T-cell assays is essential to confirm their immunogenic potential and interaction with the immune microenvironment, especially given the limited direct evidence linking these genes to neoantigen formation.

Of the nine putative proteomic biomarkers studied in an independent cohort using IHC, CDK4, ADAM10, SCAI, and DDX11 seem promising starting points for further investigation of their role in melanoma routine diagnostics. Besides the IHC expression intensity, the difference in spatial expression in melanoma cells and the stroma/TME demonstrated the complexity of the tumor-stroma relationship.

CDK4 is a well-known cancer target that regulates cell cycle and proliferation. In melanoma, mutations and dysregulation are commonly seen in CDK4 and proteins in its pathways, and several candidate drugs are in clinical trials [125,126,127]. Here, high stromal CDK4 levels in primary tumors were correlated with improved survival, while a significant increase in CDK4 levels was observed in metastatic tumor cells. It has been shown that CDK4/CDK6 inhibitors could have an essential role in tumor growth [128]. Their results suggested that in vitro, the CDK4/CDK6 inhibited fibroblasts can induce genotype-dependent tumor cell proliferation and prolonged inhibition of senescent cells in the TME. In agreement with our findings, this study raises some questions about the hidden stromal effect of the CDK4 pathways.

ADAM10 is a metalloproteinase that, by cleaving the ectodomains of transmembrane proteins, has a widespread effect on cancer cells and their stromal counterparts [129,130]. Our phosphoproteomic data point to a link between the conserved phosphosite in ADAM10 and T719 and improved survival, mirrored by protein expression in the primary melanoma cohort. Analogous to the T735 of ADAM17 phosphorylated by MAP kinases [131,132,133], T719 in ADAM10 may activate the protease.

SCAI is a highly conserved protein that acts on the RhoA–Dia1 pathway to regulate invasive cell migration [134]. SCAI protein has a debated role in cancer prognostics, correlating with a better outcome in breast and lung cancers [135]. We found that high expression of SCAI in melanoma cells and the TME was associated with better outcomes in primary melanoma.

The helicase DDX11 has a role in chromatid cohesion, influencing proliferation and melanoma cell survival [136]. DDX11 expression has been reported upregulated during melanoma progression [137,138,139], which mirrored our proteomic and IHC analysis results.

## 5. Conclusions

The findings suggest that our tumor stratification approach could improve clinical tumor characterization, potentially leading to more informed medical decisions. This study paves the way for developing models to investigate drug response-resistance dynamics, tumor dormancy, and immune responses, potentially uncovering new therapeutic targets [140]. The research benefits both the scientific and clinical communities. It offers an integrated multi-omics dataset for basic scientists, providing a valuable resource for generating new hypotheses.

## 6. Data and Code Availability

MS global proteomic data and phosphoproteomic (PXD035206) data have been deposited and are publicly available as of the date of publication.Transcriptomic data are publicly available https://doi.org/10.18632/oncotarget.3655Histology images reported in this paper will be shared by the lead contact upon request.All original code has been deposited to the GitHub repository: https://github.com/rhong3/Segundo_Melanoma (accessed on 20 November 2019) and https://github.com/bszeitz/MM_Segundo (accessed on 15 October 2021), and is publicly available as of the date of publication.Any additional information required to reanalyze the data reported in this paper is available from the lead contact upon request.

## Figures and Tables

**Figure 1 cancers-17-00832-f001:**
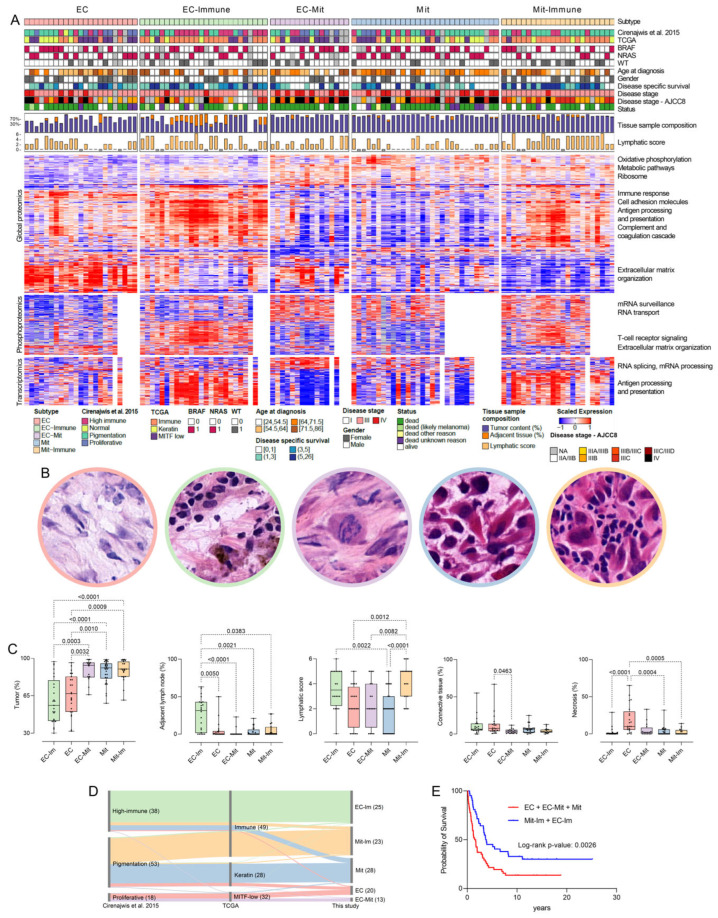
Proteomic classification of metastatic melanoma. (**A**) Overview of tumors grouped according to the proteomic subtypes established in this study, annotated with the transcriptomic classifications and clinical and histological data. Heatmaps of the most variable proteins (top 500, FDR < 0.005), phosphosites (top 1000, FDR < 0.05), and transcripts (top 500, FDR < 0.05) based on ANOVA across the five proteomic subtypes and enriched pathways annotation. (**B**) Representative histological images of the proteomic subtypes, EC (pink), EC-Im (green), EC-Mit (purple), Mit (blue), and Mit-Im (yellow). (**C**) Distribution of the annotated histological parameters, tumor content, adjacent lymph node, lymphatic score, connective tissue, and necrosis among the five proteomic subtypes (Kruskal–Wallis and Dunn’s multiple comparisons test). (**D**) Sankey diagram showing the association between proteomic (this study) and published transcriptomic subtypes. The EC-Im subtype was significantly associated with the high-immune (FDR = 0.021) and immune (FDR = 0.0095) groups. The Mit-Im subtype was significantly linked to the pigmentation (FDR = 0.021) and the immune (FDR = 0.0021) groups. In comparison, the EC-Mit was significantly associated with the proliferative (*p*-value = 0.014) and the MITF-low (FDR = 0.026) groups. The Mit proteomic subtype was significantly linked to the keratin class (FDR = 0.040) from the TCGA classification. The EC subtype was associated with the proliferative (*p*-value = 0.0244) and the MITF-low (*p*-value = 0.032) groups. (**E**) Disease-specific survival (DSS, time from surgical intervention to death or censoring) probability for patients with tumors in subtypes associated with long and short survival.

**Figure 2 cancers-17-00832-f002:**
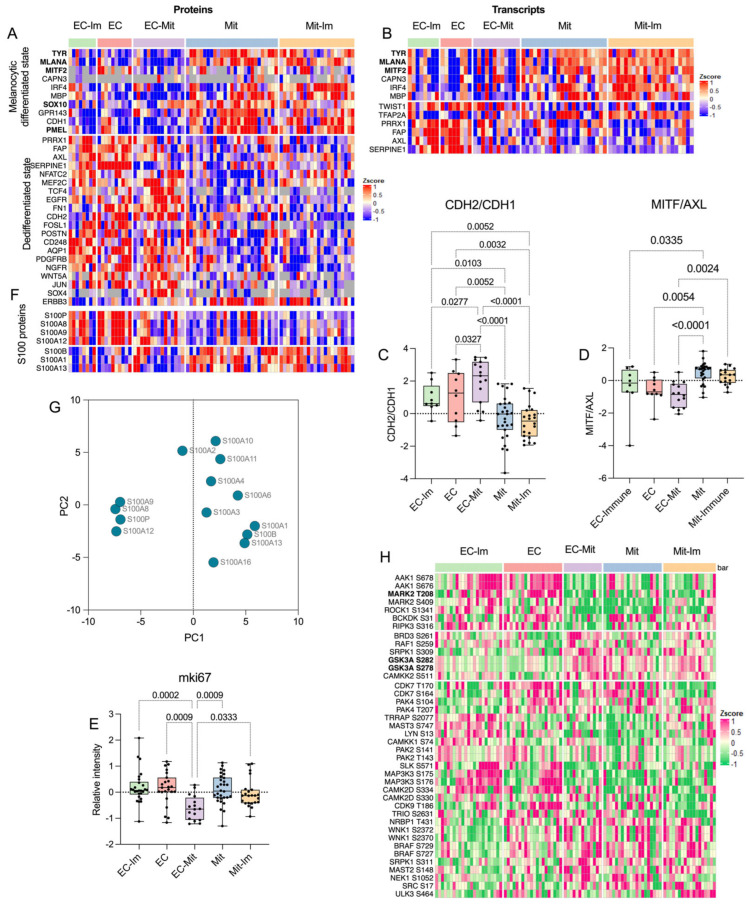
Markers of melanoma and various phenotypic states among the five proteomic subtypes. (**A**) Protein expression of melanoma markers and markers of EMT in each subtype. In bold, melanoma markers commonly used in clinical practice using IHC. (**B**) Transcript expression of melanoma markers and markers of EMT in each subtype. In bold, melanoma markers commonly used in clinical practice using IHC. (**C**) Ratio comparison of the EMT markers CDH1 and CDH2 across the proteomic subtypes. (**D**) Ratio comparison of the EMT markers MITF and AXL across the proteomic subtypes. (**E**) Protein expression of the proliferation marker mki67 across the proteomic subtypes. (**F**) S100 protein expression across the proteomic subtypes. (**G**) Principal component analysis of the S100 protein expression. (**H**) Dysregulated, activated kinases across the proteomic subtypes.

**Figure 3 cancers-17-00832-f003:**
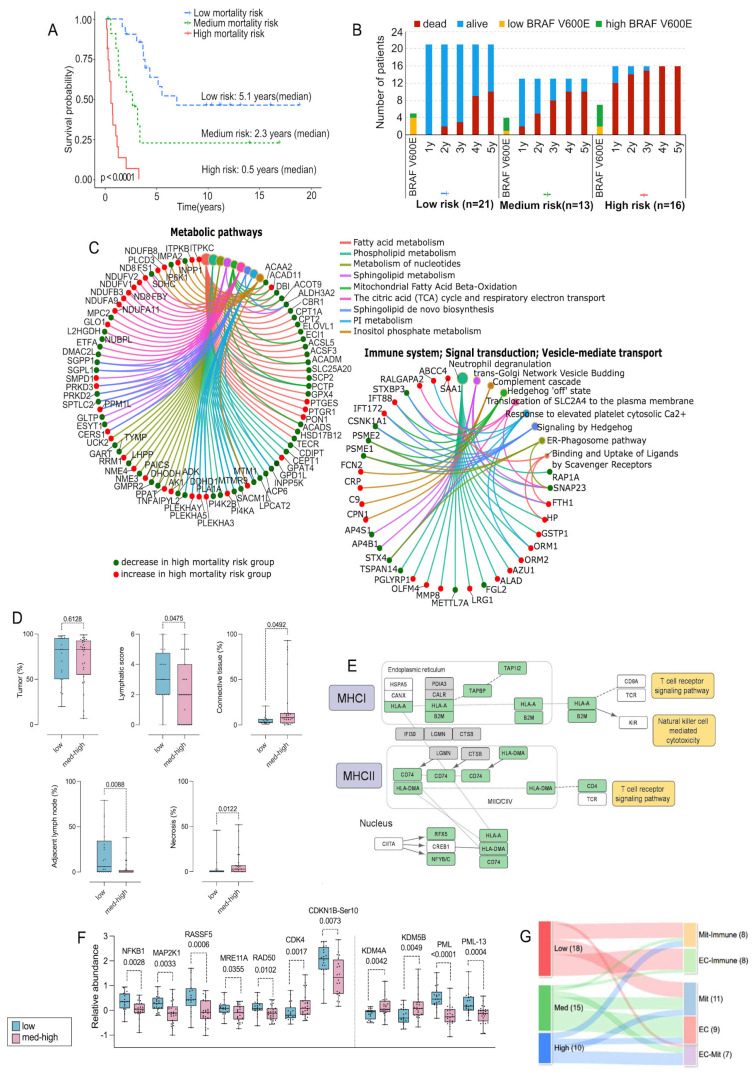
Insights from studying BRAF V600E-mutated metastases. (**A**) Kaplan–Meier curves of BRAF V600 subgroups of patients. Subgroups colored by survival probabilities: red (high risk of mortality, n = 16), green (medium risk, n = 12), and blue (low risk, n = 21). Median survival times for the three groups are shown. (**B**) Status (alive/dead) distribution of BRAF patients after 1 to 5 years from sample collection. X-axis: patient distribution stratified according to mortality risk classification. The number of patients in which levels of mutated BRAF V600E protein could be quantified is indicated in yellow (low) and green (high). (**C**) Proteins from the most significant pathways enriched in the patients with BRAF mutation. Green and red indicate the decrease and increase of protein expression, respectively, in the high-mortality-risk group. (**D**) Histological features linked to BRAF mortality groups. (**E**) Significantly upregulated proteins (green) of the antigen processing and presentation pathway in the low-mortality-risk group compared to the medium-high-risk group. Identified proteins are shown in (grey). (**F**) Proteins and phosphorylation sites linked to cellular senescence and their expression patterns between the low- and medium-high mortality-risk groups. (**G**) Association between BRAF mortality groups and the five proteomic subtypes.

**Figure 4 cancers-17-00832-f004:**
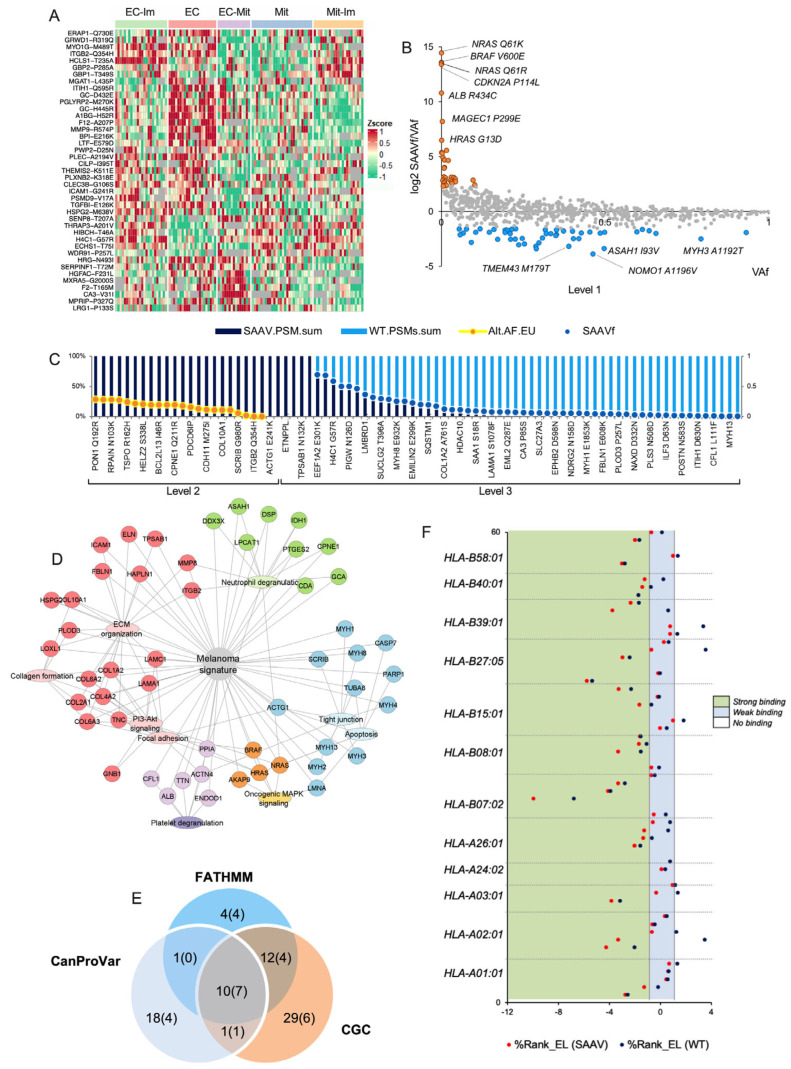
The landscape of SAAVs in melanoma. (**A**) Differentially expressed SAAVs across the proteomic subtypes. (**B**) Level 1 of the melanoma-associated SAAV signature defined in this study, with over-(orange) and under-(blue) represented SAAVs from what is expected based on SAAVf and Vaf ratio. (**C**) Proteomic evidence (#PSMs of wild-type and SAAV peptides) and genomic data (VAf) for Levels 2 and 3 of melanoma-associated SAAV signature defined in this study. (**D**) Melanoma-associated SAAV signature interconnections between corresponding proteins/genes and pathways, based on enrichment analysis using KEGG and the Reactome databases. (**E**) Overlap of the SAAVs and genes annotated by CanProvar and CGS or predicted by FATHMM as implicated in cancer. The numbers in parenthesis represent the melanoma-associated SAAV signature covered by these predictors. (**F**) Predicted affinity SAAV-neoantigen candidates per HLA, which ranked better than wild-type counterparts using NetMHC.

**Figure 5 cancers-17-00832-f005:**
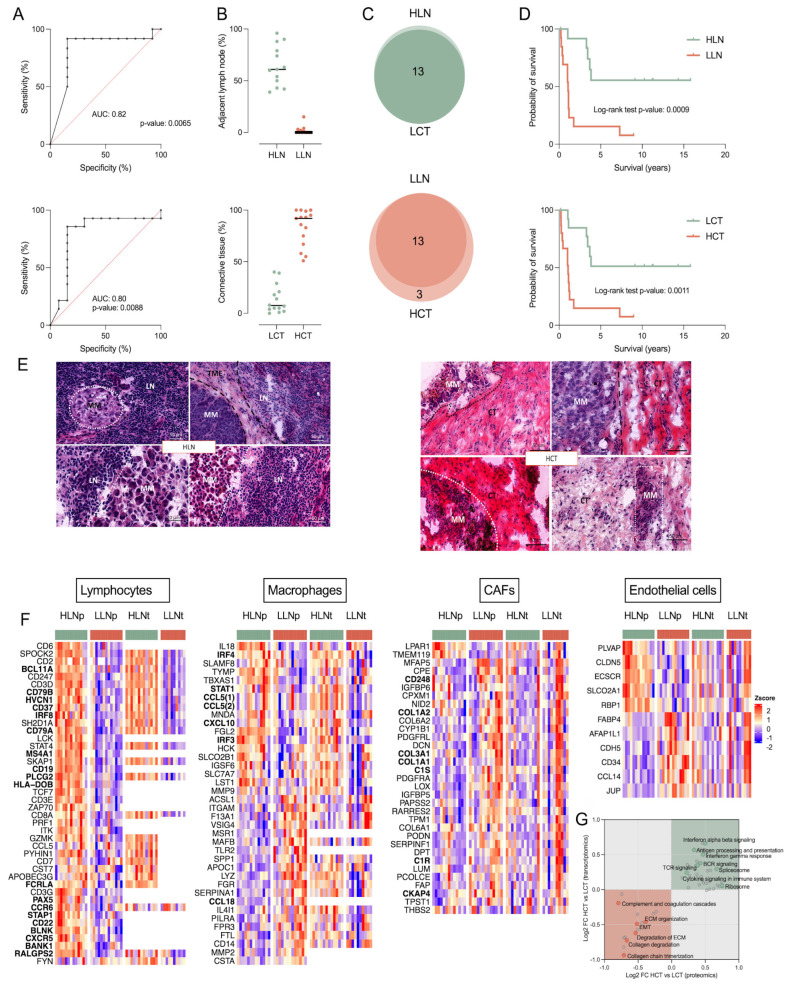
The tumor microenvironment composition is an independent prognostic marker in melanoma lymph node metastases. (**A**) ROCs of tumor-associated adjacent lymph node (top) and connective tissue (bottom) for DSS < or ≥3 years. (**B**) Adjacent lymph node (top, cut-off = 27%) and connective tissue (bottom, cut-off = 45.5%) content in the TME for the subgroups of samples generated from the ROC analysis. (**C**) Venn diagram of the patient overlap between the HLN and LCT groups and between the LLN and HCT groups. (**D**) DSS probability for patients with tumors grouped based on their content, HLN and LLN (top) or HCT and LCT (bottom). (**E**) Histological images from different tumor areas in the HLN and HCT groups. (**F**) Cell-specific signatures at protein (HLN.P and LLN.P) and transcript level (HLN.T and LLN.T) for the HLN and LLN groups. The displayed markers were significant in either the proteomic or transcriptomic analyses (*t*-test *p*-value < 0.05). Bold indicates significance in both. (**G**) 2D enrichment analysis displaying significant pathways (FDR < 0.001) commonly dysregulated on the proteomic and transcriptomic levels between the HLN and LLN groups.

**Figure 6 cancers-17-00832-f006:**
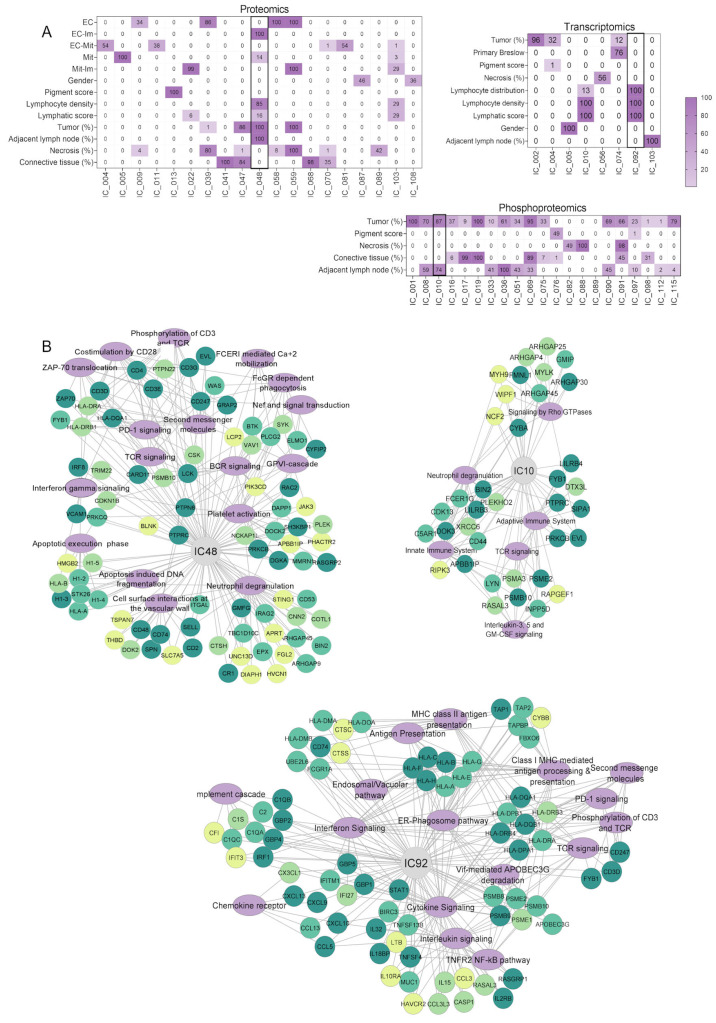
ICA connects pathways with clinical and histopathological features. (**A**) Percentage of significant correlations between independent components (ICs) and clinical and histological features (*p*-value < 0.00001) for each omics dataset. IC showing the highest overall correlation percentage in each dataset is highlighted with a black contour. (**B**) Interconnections between proteins with an IC score > 2 and pathways based on enrichment analysis using the Reactome database (FDR < 0.05) for the ICs 48 (proteomics), 10 (phosphoproteomics), and 92 (transcriptomics). The green color scale of the protein nodes indicates their quartile (dark green corresponds to the highest protein score).

**Figure 7 cancers-17-00832-f007:**
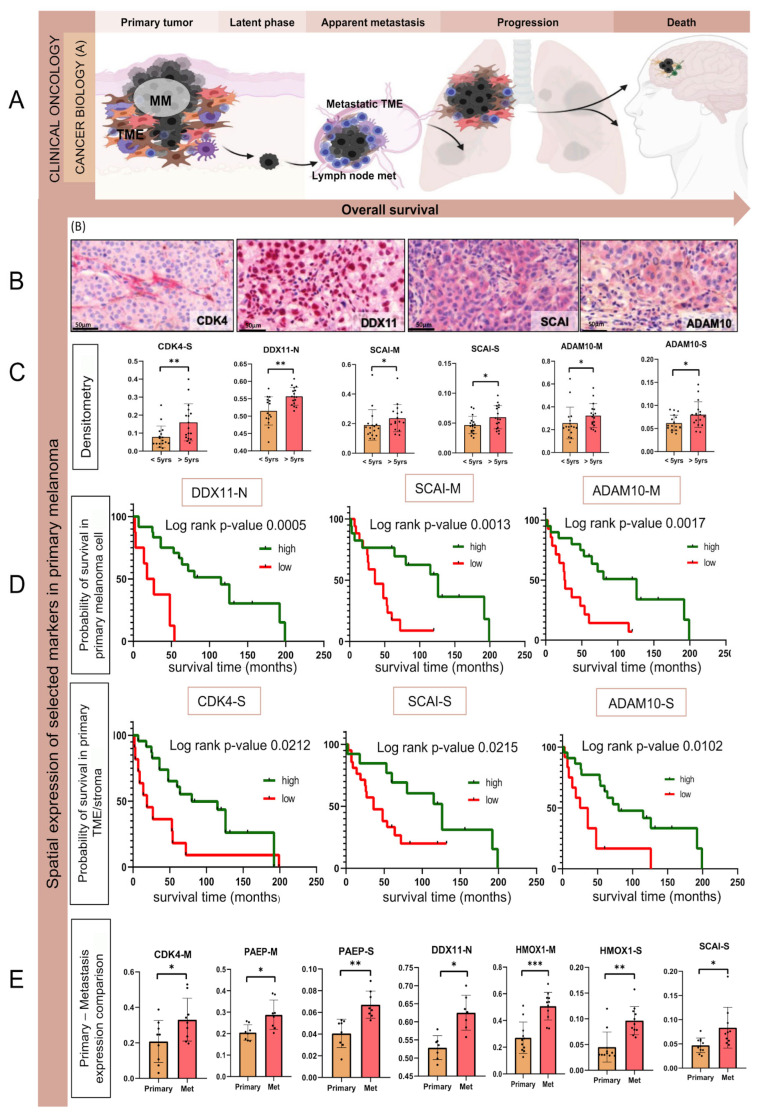
Association of protein marker expression with survival as the disease progression. (**A**) Visualization of different phases of melanoma showing the relationship between cancer biology and its clinical impact on survival. (**B**) Representative IHC staining images of markers expression in melanoma and stromal cells of primary tumors. (**C**) Significant differences found in marker expression associated with OS. (**D**) Kaplan–Meier survival analyses displaying OS rates for patients in association with high (green) and low (red) expression of the markers in melanoma and stroma cells. (**E**) Significant differences of the protein markers in pairs of matching primary melanomas and corresponding lymph node metastases. *, **, and *** above the bar plots indicate significant *p*-values of <0.05, <0.001, and <0.0001, respectively.

## Data Availability

The mass spectrometry proteomics data have been deposited in the ProteomeXchange Consortium via the PRIDE partner repository with the dataset identifier PXD035206 and are publicly available as of the publication date. The transcriptomic data can be accessed at https://doi.org/10.18632/oncotarget.3655. Histology images reported in this paper will be shared by the lead contact upon request.

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
