# Peer review of "Proteogenomic Profiling of Treatment-Naïve Metastatic Malignant Melanoma"

_cancers, 2025, doi:10.3390/cancers17050832_

Round 1
Reviewer 1 Report (Previous Reviewer 1)
Comments and Suggestions for Authors
Dear authors, I accept your reply and corrections.
greetings
This manuscript is a resubmission of an earlier submission. The following is a list of the peer review reports and author responses from that submission.
Round 1
Reviewer 1 Report
Comments and Suggestions for Authors
Dear authors,
I am sorry but I am not agree to publish this manuscript actually.
According to my experience is very dangerous to compare so ancient sample with more recent one. We can not be sure of storage condition and origin of data to compare these to ones of today.
I suggest you to use your big data for select some molecules of interest in melanoma management and purpose it for a new manuscript.
Greeting instead for you figures, I think that are original !
Thank you and good luck !
Author Response
Thank you for recognizing the originality of our figures. However, we respectfully disagree with most of your comments and would like to provide further clarification.
The primary aim of this study is to investigate a unique cohort of treatment-naïve patients who have not undergone prior targeted or immune checkpoint therapies. This is the reason, why we use retrospective samples. All tissue samples were collected under a strict workflow with electronic surveillance and stored immediately at -80°C in the Lund Biobank, a facility with decades of experience in managing high-quality patient samples. The biobank operates under stringent quality control measures, ensuring consistent sample handling, storage, and processing that allows for tracking at any part of the sample processing.
- Marko-Varga G. “BioBanking - The Holy Grail of Novel Drug and Diagnostic Developments?”, J. Clin. Bioinformat., 1, 14, 2011
- Végvári A., Welinder C., Lindberg H., Fehniger T. E., Marko-Varga G., ”Biobank resources for future patient care: developments, principles and concepts”, J. Clin. Bioinformatics., 2011, 1, 24
- Malm J., Végvári A., Rezei M., Upton P., Danmyr P., Nilsson R., Steinfelder E., Marko-Varga G., “Large Scale Biobanking of Blood – The importance of high density sample processing procedures”, J. Proteomics., 2012, 76, 116-124
- Malm J., and Marko-Varga G., in Ed. Marko-Varga G., Genomics and Proteomics for Clinical Discovery and Development, Vol 4, chapter 12: “The Role of Proteomics in the Development of Personalized Medicine and Large Scale Biobanking”, Springer Verlag 2014, pp 207-215
The Lund Biobank has collaborated extensively with major pharmaceutical companies, including AstraZeneca, and serves as a trusted partner for clinical drug studies from 10 hospitals in Southern Swedish healthcare (Region Skåne), within the BIG3 study focusing on lung cancer, cardiovascular diseases, and COPD (https://kliniskastudier.se/forum-soder/regionala-forskningsprojekt/big3 and https://snd.se/sv/catalogue/dataset/ext0129-1) . The track record is; more than 3 million samples from an 8,000 patient study has been processed with an impeccable safety record, using real-time tracking and electronic temperature control systems. As a further evidence of the preserved quality of these samples, we have expression patterns from this study, aligning with the output from prospective Melanoma studies we undertake.
In addition, our results closely align with the histopathological assessment conducted on the same tumor tissues selected for the proteomic study. Specifically, we observed expected patterns in EMT-like markers in differentiated and dedifferentiated melanomas, which are consistent with well-established findings from both historical and recent melanoma literature.
These factors underscore the robustness and reliability of our samples, validating their use in this study.
Reviewer 2 Report
Comments and Suggestions for Authors
Here the authors addressed the histopathology-driven proteogenomic landscape of 142 treatment-naïve metastatic melanoma samples and report five proteomic subtypes that integrate the immune and stroma microenvironment components, and associate with clinical and histopathological parameters, providing foundations for an in-depth molecular classification of melanoma.
I commend the authors for their detailed analysis and reporting. The manuscript is clearly written in professional, unambiguous language.
Author Response
Thank you for your positive feedback and for recognizing the effort we put into the analysis and reporting. We greatly appreciate your acknowledgment of the clarity and professionalism in the manuscript
Reviewer 3 Report
Comments and Suggestions for Authors
Authors have conducted a proteogenomic profiling study of treatment-naïve metastatic malignant melanoma, focusing on histopathology-driven analysis, proteomic subtypes, BRAF V600 mutations, tumor microenvironment, and single amino acid variants (SAAV). It’s a well written article with methodology and results discussed properly. Some suggestions to further improve the paper are:
1. Section 2.9.1 should be rewritten as sentences in a paragraph.
2. Kindly include the IHC slide pictures of all the 6 proteins as a new figure.
3. Please include a table listing the gene/protein signatures classified into levels 1-3 as results discussed for section 2.9.10.
4. The number of reference citations can be reduced if possible.
Author Response
1. We agree. Following the editor’s suggestion the detailed methods section was moved to the supplementary materials. In the current version of the supplementary materials you may find on page 9 under Statistical analysis of omics data and Sample exclusion all the information rewritten as sentences in a paragraph (see below).
“For consensus clustering and subtype analysis, samples with >30% tumor content (n=118) were included, while the analysis of phenotype-switching markers was restricted to samples with >70% tumor content (n=83) (Figure S1J). Independent component analysis (ICA) was performed on samples with >30% tumor content, and BRAF V600 mutation analysis was conducted on n=49 samples. Tumor microenvironment (TME) analysis was limited to samples with <50% tumor content (n=29). Survival analyses were carried out on samples with >30% tumor content, including outlier and Cox regression. Sample MM-SEG-0113 was excluded from all analyses due to technical issues.”
-
This suggestion is closely related to Reviewer #5 (comment #4). Therefore, we generated an additional figure in the supplementary information (Figure S7), which displays the IHC staining for all these proteins, annotated to indicate melanoma cells and stromal cells."
The IHC analysis involves four proteins, and for two of them, the protein expression was evaluated in both melanoma and stromal cells, resulting in six IHC protein expression analyses in total.
For a better understanding and following the suggestion of Reviewer #5 (comment #4) we include the following text at the end of the legend of figure 7:
“DDX11-N, SCAI-M, SCAI-S, ADAM10-M, ADAM10-S, and CDK4-S denote the IHC expression of proteins in specific cell types or cellular compartments. 'S' indicates expression in stromal cells, 'M' refers to expression in melanoma cells, and 'N' represents nuclear protein expression.
- Thanks for pointing this out. In the current version of the manuscript, paragraph #5 of the Results section 3.7 (Melanoma-associated single amino acid variants (SAAVs)) you may read: “….melanoma-associated SAAVs classified into levels 1, 2, and 3, according to genomic data and proteomic evidence (Figure 4B and 4C and Table S5A)”
The supplementary table S5A provides a comprehensive list of all SAAVs identified in this study. Within this table, the field titled 'Melanoma Signature Level' indicates whether each specific SAAV and its corresponding gene are part of the gene/protein signatures classified into levels 1-3.
- In current version of the manuscript we reduced the number of reference citations from 183 to 141. The majority of them were moved to supplementary information.
Reviewer 4 Report
Comments and Suggestions for Authors
The MS is really interesting. Majority of the experimental data supports authors claims. However there are few major concerns that needs to be addressed.
In Fig-2 Legend needs to be corrected. Figure alphabetical order is different from legend alphabetical order.
Author Response
Thank you for your feedback. In the current version of the manuscript the figure and the legend have been corrected.
Reviewer 5 Report
Comments and Suggestions for Authors
Kuras et. al. reported a multi-omics analysis on melanoma samples. They proposed a signature-based category and applied the model to the mortality risk assessment. This is a comprehensive manuscript.
Some points to consider:
1. Materials and Methods section 2.1 contains demographic summary of patients, which can be moved to the Results section, in the form of one table per cohort like Figure S1B.
2. Materials and Methods section 2.9.1 Sample exclusion is kind of confusing. For example, does
"subtype: >30% tumor content" (line 391) mean that n=118 patients were excluded from the original dataset? A flow chart similar Figure SA could be used to explain how the exclusion criteria work.
3. In terms of proteomic signatures, which set, Figure 1A or Figure S2B, is more representative and more relevant to diagnosis?
4. For better understanding, melanoma (M) and stromal (S) parts could be annotated on H&E images (Figure 7B). In addition, cell type specific expression like "DDX11-N", "SCAI-M", "SCAI-S" need to be explicitly defined.
5. Some error need to be fixed. Figure 2 has two panel G labels. Please clarify the correct ones. Figure 2H (line 891) does not exist.
Author Response
Thank you for your positive feedback. We appreciate your recognition of the manuscript's comprehensiveness.
- Thank you for this suggestion. The demographic information in the Materials and Methods section for both the discovery cohort and immunohistochemistry cohort was turned into summary tables for each cohort. A detailed description of each patient sample can be found in Tables S1A and S1B for the discovery and immunohistochemistry cohorts, respectively.
- Thank you for this comment. For a better readability the sample exclusion criteria in section 2.9.1 were converted into a paragraph. Notice that following the editor’s suggestion this section was moved to supplementary information (page 9 under Statistical analysis of omics data and Sample exclusion) (see also below).
A flowchart was also added to better explain the different criteria for the different analyses (Figure S1J).
“For consensus clustering and subtype analysis, samples with >30% tumor content (n=118) were included, while the analysis of phenotype-switching markers was restricted to samples with >70% tumor content (n=83) (Figure S1J). Independent component analysis (ICA) was performed on samples with >30% tumor content, and BRAF V600 mutation analysis was conducted on n=49 samples. Tumor microenvironment (TME) analysis was limited to samples with <50% tumor content (n=29). Survival analyses were carried out on samples with >30% tumor content, including outlier and Cox regression. Sample MM-SEG-0113 was excluded from all analyses due to technical issues.”
- As the reviewer correctly noted, our study outlined two sets of proteins to characterize the proposed melanoma subtypes. Figure 1A presents the top 500 differentially expressed proteins (FDR-based) across subtypes, summarizing the core biological pathways and processes that define these subtypes. While this dataset captures the broader biological context, the proteomic signature (Figure S2B) derived from Independent Component Analysis (ICA) may hold greater translational value for clinical diagnostics. This signature is based on the top 10 contributing proteins for each independent component significantly associated with the proteomic subtypes. As a result, it provides a more concise protein panel that, following rigorous validation, could serve as potential biomarkers to distinguish between the melanoma subtypes.
Consequently we have added the following text into the discussion section:
“Our findings suggest that melanoma subtypes can be distinguished using the MS-based abundance of routinely employed histopathologic protein markers such as MITF, PMEL (HMB45), MLANA, and specific members of the S100 protein family. The S100 proteins exhibit selective and coordinated regulation across subtypes, linked to melanoma phenotype switching, as observed in the subtype signatures identified through Independent Component Analysis (ICA). Altogether, the histopathologic markers and ICA-derived protein signatures, following rigorous validation, may serve as potential diagnostic biomarkers for patient stratification.”
-
This suggestion is closely related to Reviewer #3 (comment #2). Therefore, we generated an additional figure in the supplementary information (Figure S7), which displays the IHC staining for all four proteins, annotated to indicate melanoma cells and stromal cells."
We include the following text at the end of the legend of figure 7:
“DDX11-N, SCAI-M, SCAI-S, ADAM10-M, ADAM10-S, and CDK4-S denote the IHC expression of proteins in specific cell types or cellular compartments. 'S' indicates expression in stromal cells, 'M' refers to expression in melanoma cells, and 'N' represents nuclear protein expression.”
- The figure 2 and the legend have been corrected.